# Prevalence of Obesity and Severe Obesity among Professionally Active Adult Population in Poland and Its Strong Relationship with Cardiovascular Co-Morbidities-POL-O-CARIA 2016–2020 Study

**DOI:** 10.3390/jcm11133720

**Published:** 2022-06-27

**Authors:** Anna Rulkiewicz, Iwona Pilchowska, Wojciech Lisik, Piotr Pruszczyk, Michał Ciurzyński, Justyna Domienik-Karłowicz

**Affiliations:** 1LUX MED, Postępu 21C, 02-676 Warsaw, Poland; anna.rulkiewicz@luxmed.pl (A.R.); iwona.pilchowska@luxmed.pl (I.P.); 2Department of Psychology, SWPS University of Social Sciences and Humanities, 03-815 Warsaw, Poland; 3Department of General and Transplantation Surgery, Medical University of Warsaw, 02-014 Warsaw, Poland; wojciech.lisik@wum.edu.pl; 4Department of Internal Medicine and Cardiology, Medical University of Warsaw, 02-005 Warsaw, Poland; piotr.pruszczyk@wum.edu.pl (P.P.); michal.ciurzynski@wum.edu.pl (M.C.)

**Keywords:** BMI index, professionally active adult population, cardiovascular diseases, obesity

## Abstract

For several decades, a steady increase in the percentage of overweight and obese people has been observed all over the world. There are many studies available in the literature emphasizing the relationship of overweight and obesity with the occurrence of other diseases. The aim of this study is to characterize the prevalence of obesity and severe obesity, as well as their changes over time, among professionally active adults who underwent occupational medicine examinations in Poland in 2016–2020, for the POL-O-CARIA 2016–2020 study. In total, the results of 1,450,455 initial, control and periodic visits as part of the occupational medicine certificate were analyzed. Statistical calculations were performed with the use of IBM SPSS Statistics 25. In both groups (men/women), a significant decrease was observed every year for people who had normal body weight. In addition, the tendency to increase in people with I and III degrees of obesity was more strongly observed in the male group. A significant relationship was also observed between BMI categories and the occurrence of all analyzed comorbidities: hypertension, type 2 diabetes, lipid disorders and coronary artery disease (chi^2^ (70) = 12,228.11; *p* < 0.001). Detailed results showed that in the group of patients diagnosed with hypertension or lipid disorders, significant differences were observed between all groups; it turned out that as the BMI level increased (I, I, III), there was an increase in the percentage of occurrence of hypertension (38.1%, 41% and 45.3%, respectively) and type 2 diabetes (3.2%, 4.6% and 5.8%, respectively) (*p* < 0.001). Our analysis indicates that the prevalence of adult obesity and severe obesity will continue to increase nationwide, with an accompanying large increase in comorbidities.

## 1. Introduction

For several decades, a steady increase in the percentage of overweight and obese people has been observed all over the world. More and more countries declare problems with controlling this epidemic. This disease affects children as well as adults [1]. According to the WHO definition, obesity is abnormal or excessive accumulation of fat that negatively affects health. Obesity is diagnosed when the BMI level exceeds or is equal to 30 kg/m^2^ [2]. According to data published by WHO in 2014, the percentage of people with obesity in Poland is 25.1% [3]. The main cause of obesity is a long-term imbalance between the amount of calories consumed and the body’s demands [4]. Diet, lifestyle and genetics have a significant influence on the occurrence of obesity [5].

### 1.1. Obesity and Comorbidities

There are many studies available in the literature emphasizing the relationship of overweight and obesity with the occurrence of other diseases. A meta-analysis carried out in 2015 showed that each increase in weight by 5 kg significantly increases the risk of developing post-menopausal breast (11%), endometrial (39%), ovarian cancer (13%) and male colon cancer (9%) [6]. Cohort studies conducted in Europe (Austria, Norway, Sweden) under the Me-Can 2.0 program showed that overweight people up to 40 years of age significantly increase their chance of developing endometrial, male renal cell and male colon cancer [7]. Obesity is a chronic and metabolic disease; therefore, it affects the occurrence of cardiovascular diseases [8]. It affects the structural and functional changes in the cardiovascular system, e.g., causing decreased cardiac output, increased left ventricular mass and wall thickness [9]. The association of obesity with hypertension, coronary artery disease and diabetes is also scientifically confirmed [10,11].

### 1.2. The Global Obesity Epidemic

For several decades, a steady increase in the percentage of overweight and obese people has been observed. Most countries in the world are affected. Current reports show that more people worldwide die from overweight and obesity than from underweight [12]. Obesity is the main reason for the development of NCDs (non-communicable diseases), which since 2010 have been responsible for 86% of deaths and 77% of other diseases in Europe. Over the past 40 years, there has been a sharp increase in the percentage of people with obesity; since 1975, the percentage of people with obesity has increased from 1% to 6–8%. Women saw an increase from 6% to 15%, while men increased from 3% to 11% [13]. There are four levels of obesity, distinguished on the basis of an analysis of the 30 most populous countries in the world [14]:-Level 1—characterized by a higher prevalence of obesity in women than men (more often in adults than children) and in people with a higher socioeconomic status. This level is most commonly observed in South Asia and Sub-Saharan Africa.-Level 2—at this stage there is a significant increase in obesity in the adult group and a decrease in children. The importance of gender and socioeconomic status is not as clear as in level 1. This stage is most often observed in Latin America and the Middle East.-Level 3—the most characteristic for the inhabitants of Europe. A higher obesity rate is observed more often in the group of people with a low socioeconomic status, but it is worth noting the increase in the percentage of obese people in the group of women with high economic status and in the group of children.-Level 4—there are few countries classified to this stage. It is characterized by a decrease in the prevalence of obesity. The research results do not allow for an unequivocal determination of the relationship between the prevalence of obesity and gender and socioeconomic status.

The two regions with a dynamic increase in obesity are North America and Europe [14].

### 1.3. Actions to Reduce the Obesity Epidemic

Obesity is a multidimensional disease that affects many spheres of life. Hence, it is advisable to provide long-term support for patients suffering from this disease. Current activities aimed at controlling and reducing obesity in society focus on the analysis of the occurrence of civilization diseases, followed by body weight. It seems important to focus on the many dimensions of the fight against obesity (diet, physical activity, changes in behavior), which will translate into an improvement in the quality of life. Attention is also paid to the growing interest in surgical methods of obesity treatment [15]. In Europe, the prevalence of overweight increased from 48% in 1980 to 59.6% in 2015; in the case of obesity, the incidence increased from 15.5% in 1980 to 22.9% in 2015. Moreover, a lower probability of obesity was reported among women (in the group of people aged 20 to 44); an inverse relationship was observed in the group of people over 45 years of age.

### 1.4. Obesity and Professional Activity

Work is recognized as a source that may influence overweight and obesity [16]. Employers take measures to promote healthy eating habits and increase physical activity among employees [17]. Employers increasingly organize free fruits and vegetables in their offices for their employees [18]. The factors that may increase the chance of overweight and obesity in the workplace are sedentary work, stress and sleep problems [19]. Office work and sedentary work increase the likelihood of obesity among employees [20]. The research by Shields and Tremlay (2008) confirmed the existence of a positive relationship between obesity and spending free time sitting (e.g., while using a computer) [21]. There are also several studies that do not confirm the relationship between sedentary work or leisure activities and the prevalence of overweight and obesity [22]. An important factor associated with overweight and obesity is also stress experienced in the workplace [23].

### 1.5. Aim of the Study

The aim of the study is to characterize professionally active adults who underwent occupational medicine examinations in Poland in 2016–2020. Due to the exploratory nature of the research, the article did not put forward research hypotheses. Instead, research questions were asked that define the main subject of the analysis: how the intensity of obesity changes over time and how it coexists with other diseases [24].

## 2. Materials and Methods

The article analyzes the results of the POL-O-CARIA 2016–2020 study, concerning adults who are professionally active and visited in the years January 2016–April 2020 as part of occupational medicine. The data for analysis was provided by the LUX MED Group. In total, the results of 1,450,455 initial, control and periodic visits as part of the occupational medicine certificate were analyzed. During the study, sex, age, province of residence, information on the period of the issued medical certificate and data contained in the medical history (subjective assessment of health, smoking) were controlled. Detailed characteristics of the studied patients are presented in Appendix A.

For several decades, a steady increase in the percentage of overweight and obese people has been observed. For this reason, it seems extremely important to monitor the prevalence of obesity in individual social groups. The study of professionally active adults is important for several reasons. It is important to monitor the health condition and forecast the occurrence of specific civilization diseases in a given society. The occurrence of certain diseases (e.g., obesity, hypertension, diabetes) translates into shorter medical certificates enabling employment.

### Statistical Analysis

Statistical calculations were performed with the use of IBM SPSS Statistics 25 [25]. Percentage and number of occurrences were used to analyze qualitative data, while the following were used to characterize qualitative data: mean (M), standard deviation (SD), median, skewness, kurtosis, and the minimum and maximum values. Significant statistical results were considered to be those where the probability of making a type I error was lower than 5% (*p* < 0.05). The following were used for statistical calculations: chi-square analysis in cross tables (Bonferroni correction was used to test column proportions) and one-way analysis of variance (Scheffe’s post hoc test was used for mean comparisons). The charts were made in the R program [26].

## 3. Results

### 3.1. Information on BMI

It was observed that, with successive years of measurement, the percentage of overweight and obesity (regardless of degree) increased, while the percentage of people with normal body weight significantly decreased. Detailed results are presented in Table 1.

In the case of BMI, it was shown that the longest medical certificates for professionally active Poles were received by people with normal body weight, which equaled about 34 months. In the group of people with overweight and obesity, it was observed that, along with the degree of obesity, the average number of months in which the decision was issued decreased significantly (see Table 2). According to doctor decision, patients with overweight received medical certificates for work for an average of about 30 months. The ability to work was significantly worse in patients with obesity of the I degree (about 28 months), obesity of the II degree (about 27 months) and obesity of the III degree (almost 26 months); the details are presented in Table 2.

### 3.2. Patient Characteristics Depending on the BMI Level

Chi-square analysis showed that, for both women and men, similar trends were observed regarding the dynamics of occurrence of individual BMI categories. In both groups, a significant decrease was observed every year for people who had normal body weight. In addition, the tendency to increase in people with I and III degrees of obesity was more strongly observed in the male group (see Table 3).

Significant differences, regardless of the year of measurement, were observed using a one-way analysis of variance for the age and time for which the measurement was issued (in both cases, the significance of differences between the groups was *p* < 0.001). The exact results are discussed below.

In the case of patients’ age, post hoc analysis with Scheffe’s correction showed that only between people with II and III degrees of obesity were there no differences for the average age; in other cases, the significance of differences between individual groups was *p* < 0.001. The highest average age was observed for people with obesity, while the lowest was observed for people with underweight or normal weight (see Figure 1).

Patients with normal body weight most often occurred in the group under 35 years of age, while the percentage of people with obesity (especially I degree) increased significantly in each age category (see Table 4).

Table 5 presents the same data by changing the percentage to the BMI category. The obtained results showed that, together with the higher BMI level, the percentage of people under 35 years of age decreased in each group. In the case of people aged 35–69, it was obtained that they were more often classified into the group with obesity or overweight, compared to groups with normal body weight.

When analyzing the time periods for issuing a medical certificate, significant differences between the groups were also observed. A linear trend was obtained showing that, along with the BMI level, the average number of months of the issued decision decreased. In addition, post hoc analysis with Scheffe’s correction showed that significant differences were observed between all BMI categories. Detailed results are presented below (see Figure 2).

Patients with normal weight or underweight were less likely to smoke than overweight or obese patients. This relationship was observed regardless of the year of measurement (see Figure 3).

The relationship between BMI categories and subjective health assessment was also examined. It was found that people who subjectively assessed their own health as good, less often than people who assessed their health as very good, were classified into the group of people with normal body weight. An inverse relationship was obtained for overweight and obese people. Detailed results are presented below Table 6.

Table 7 shows the relationship between selected diseases and BMI categories. A significant relationship between variables was obtained (*p* < 0.001). The most pronounced differences were observed for hypertension (with increasing BMI level, the percentage of people with this disease increased), and for lipid disorders and type 2 diabetes.

### 3.3. BMI and Observed Comorbidities

A significant relationship was also observed between BMI categories and the occurrence of comorbidities (chi^2^ (70) = 12,228.11; *p* < 0.001). Detailed results showed that in the group of patients diagnosed with hypertension or lipid disorders, significant differences were observed between all groups; it turned out that, as the BMI level increased, the percentage of occurrence of a given disease increases. A comparison of all comorbidities depending on BMI level is shown in the Table 8 below.

The cross-tabulation chi-square analysis performed confirmed that there was an association between age and comorbidities (chi^2^(56) = 16,758.06; *p* < 0.001). In the case of hypertension, it was obtained that the prevalence of hypertension was more common in those aged 18–54 compared to other age groups. In addition, the prevalence of lipid disorders was significantly different in each of the age groups; a trend was observed showing that the diagnosis of this disease decreased with age. A detailed comparison of the age groups for the other diseases is shown below Table 9.

## 4. Discussion

In this study, we used data from 931,985 unique adult patients and applied an analytical approach that provided estimates of BMI trends. Analyses on the prevalence of obesity in Poland were present in previous years; however, none of them were concerning the current years, and they were not based on such a large group of patients.

Unique in our analysis is also the correlation with the average number of months for the issued medical certificate, and the correlation with the coexistence of other serious diseases, mainly of the cardiovascular system. It is very worrying that, with the increase in BMI, the ability to work is limited, and we did not include patients who, due to obesity and comorbidities, do not try to work at all.

We would like to point out that, in this very large group of patients, we have confirmed the coexistence of diseases that significantly reduce the quality of life of patients, and their coexistence clearly depends on the degree of obesity.

Our data showed that one third of the professionally active women and almost two thirds of the professionally active men are overweight or obese. This result is extremely disturbing. Moreover, we demonstrated a trend showing an increase in the phenomenon over time, which raises concerns in terms of access to medical care and the cost of medical care. The data clearly indicate that the phenomenon is not uniform in all regions of the country. In additional materials, we present unique data indicating the diversification of obesity depending on the region of the country. Although grade II and grade III obesity were once a rare condition, our findings may suggest that they will soon be the most common BMI category in the patient populations. Given that physicians are not well equipped to treat obese patients, the continuing trend will be a major challenge for healthcare as a whole.

## 5. Conclusions

Further annual assessment of the prevalence of obesity and comorbidities seems necessary to prepare the health care system for treating growing number of obese, professionally active Poles, and to take the most effective measures to inhibit the trend.

## Figures and Tables

**Figure 1 jcm-11-03720-f001:**
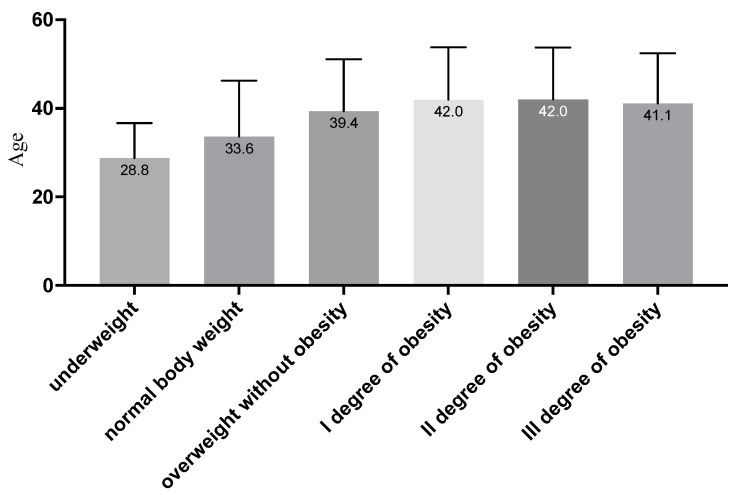
Average age depending on the BMI category (in the figure, all groups are statistically significantly different, at least at the *p* < 0.001 level; due to the number of groups compared, results for differences are not shown in the figure).

**Figure 2 jcm-11-03720-f002:**
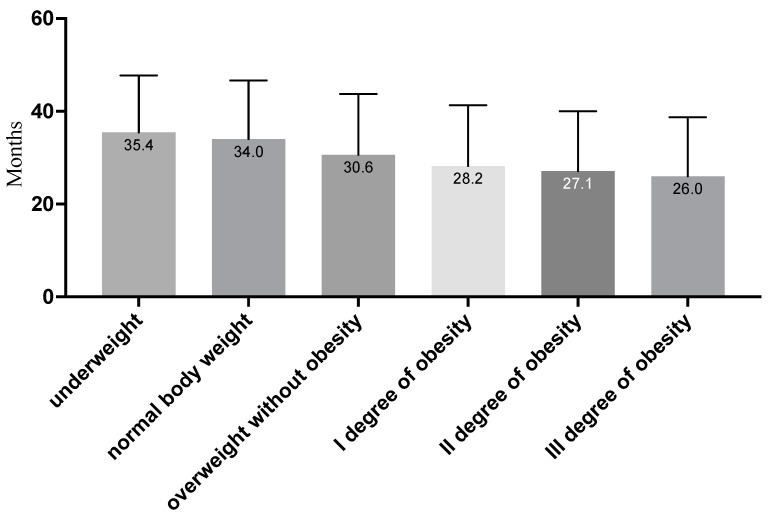
Average number of months for the issued medical certificate depending on the BMI category (in the figure, all groups are statistically significantly different, at least at the *p* < 0.05 level; due to the number of groups compared, results for differences are not shown in the figure).

**Figure 3 jcm-11-03720-f003:**
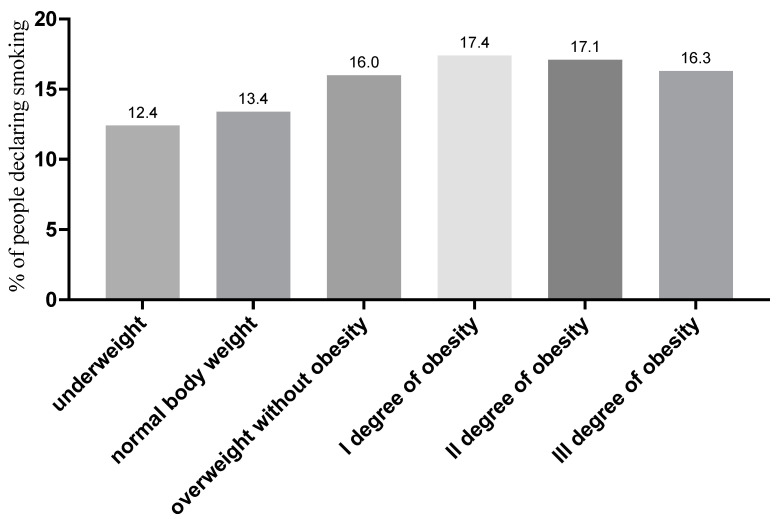
Percentage of people declaring smoking depending on the BMI category (due to the number of groups compared, results for differences are not shown in the figure).

**Table 1 jcm-11-03720-t001:** Body mass index (BMI) distribution depending on the year of measurement.

	2016	2017	2018	2019	2020	Total
underweight	3.40% _a_	3.40% _a_	3.40% _a_	3.20% _a_	2.90% _a_	3.30%
normal body weight	51.60% _a_	51.10% _a_	50.20% _b_	49.30% _b_	46.90% _c_	50.30%
overweight without obesity	31.40% _a_	31.50% _a_	32.00% _a_	32.20% _a_	33.50% _a_	31.90%
I degree of obesity	10.40% _a_	10.70% _a,b_	11.00% _b,c_	11.50% _c_	12.50% _d_	11.00%
II degree of obesity	2.50% _a_	2.60% _a_	2.70% _a,b_	2.90% _b_	3.30% _b_	2.70%
III degree of obesity	0.70% _a_	0.70% _a_	0.80% _a,b_	0.80% _a,b_	0.90% _b_	0.80%
Total	100.0%	100.0%	100.0%	100.0%	100.0%	100.0%

Each letter in subscript represents a subset of the year category whose column proportions do not differ significantly at the level of 5%.

**Table 2 jcm-11-03720-t002:** Descriptive statistics on the average number of months of the issued medical certificate depending on BMI.

	Underweight	Normal Body Weight	Overweight without Obesity	I Degree of Obesity	II Degree of Obesity	III Degree of Obesity
M	35.46	34.01	30.63	28.24	27.10	25.99
Me	36.00	36.00	36.00	24.00	24.00	24.00
SD	12.28	12.63	13.06	13.07	12.90	12.72
Skewness	−0.71	−0.57	−0.25	−0.05	0.05	0.14
Kurtosis	−0.02	−0.50	−0.80	−0.91	−0.66	−0.83
Min	0.00	0.00	0.00	0.00	0.00	0.00
Max	156.00	156.00	178.00	155.00	156.00	60.00

M—mean; Me—median; SD—standard deviation.

**Table 3 jcm-11-03720-t003:** Relationship between BMI and measurement time, as well as patients’ sex; data presented as percentage of the year of measurement ^1^.

		2016	2017	2018	2019	2020	Total
Women	underweight	6.1% _a_	6.2% _a_	6.0% _a_	5.8% _b_	5.4% _c_	6.00%
	normal body weight	64.4% _a_	63.7% _b_	62.7% _c_	62.0% _d_	60.3% _e_	63.00%
	overweight without obesity	19.6% _a_	19.9% _a_	20.7% _b_	20.9% _b_	22.0% _c_	20.40%
	I degree of obesity	7.1% _a_	7.3% _a_	7.5% _b_	8.0% _c_	8.6% _d_	7.50%
	II degree of obesity	2.1% _a_	2.2% _b_	2.2% _b_	2.4% _c_	2.8% _d_	2.30%
	III degree of obesity	0.7% _a_	0.8% _b_	0.8% _b,c_	0.8% _c,d_	0.9% _d_	0.80%
	Total	100.00%	100.00%	100.00%	100.00%	100.00%	100.00%
Men	underweight	0.8% _a_	0.8% _a,b_	0.9% _b,c_	0.9% _c_	0.8% _a,b_	0.80%
	normal body weight	39.3% _a_	38.9% _b_	38.3% _c_	37.9% _d_	35.5% _e_	38.40%
	overweight without obesity	42.7% _a,b,c,d_	42.8% _c,d_	42.7% _b,d_	42.4% _b_	43.2% _a,c_	42.70%
	I degree of obesity	13.6% _a_	14.0% _b_	14.3% _c_	14.7% _d_	15.9% _e_	14.30%
	II degree of obesity	2.9% _a_	2.9% _a_	3.2% _b_	3.3% _b_	3.7% _c_	3.10%
	III degree of obesity	0.6% _a_	0.7% _a_	0.8% _b_	0.8% _c_	0.9% _c_	0.70%
	Total	100.00%	100.00%	100.00%	100.00%	100.00%	100.00%

^1^ Each letter in subscript represents a subset of the year category whose column proportions do not differ significantly at the level of 5%.

**Table 4 jcm-11-03720-t004:** Relationship between BMI and patients’ age; data presented as percentage of age group ^1^.

	<18	18–35	35–54	55–69	>69	Total
underweight	14.3% _a_	5.1% _b_	1.4% _c_	0.5% _d_	0.6% _d_	3.30%
normal body weight	61.7% _a_	60.1% _a_	41.7% _b_	27.5% _c_	26.9% _c_	50.20%
overweight without obesity	17.5% _a_	25.6% _b_	37.8% _c_	44.8% _d_	49.9% _e_	32.00%
I degree of obesity	4.0% _a_	7.0% _b_	14.4% _c_	20.9% _d_	18.2% _e_	11.10%
II degree of obesity	1.9% _a_	1.7% _a_	3.7% _b_	5.0% _c_	4.0% _b_	2.70%
III degree of obesity	0.5% _a,b,c,d_	0.5% _d_	1.1% _c_	1.2% _b_	0.4% _a,d_	0.80%
Total	100.00%	100.00%	100.00%	100.00%	100.00%	100.00%

^1^ Each letter in subscript represents a subset of the year category whose column proportions do not differ significantly at the level of 5%.

**Table 5 jcm-11-03720-t005:** Relationship between BMI and age of patients; data presented as percentage of the BMI category ^1^.

	Underweight	Normal Body Weight	Overweight Without Obesity	I Degree of Obesity	II Degree of Obesity	III Degree of Obesity	Total
<18	0.2% _a_	0.1% _b_	0.0% _c_	0.0% _d_	0.0% _c_	0.0% _b,c,d_	0.10%
18–35	83.2% _a_	64.1% _b_	42.9% _c_	33.7% _d_	32.8% _e_	34.7% _f_	53.50%
35–54	15.0% _a_	30.5% _b_	43.3% _c_	47.8% _d_	49.3% _e_	50.5% _f_	36.70%
55–69	1.5% _a_	5.3% _b_	13.5% _c_	18.2% _d_	17.6% _e_	14.7% _f_	9.60%
>69	0.0% _a_	0.1% _b_	0.3% _c_	0.3% _c_	0.2% _c_	0.1% _b_	0.20%
Total	100.0%	100.0%	100.0%	100.0%	100.0%	100.0%	100.0%

^1^ Each letter in subscript represents a subset of the year category whose column proportions do not differ significantly at the level of 5%.

**Table 6 jcm-11-03720-t006:** Relationship between BMI and subjective assessment of health; data presented as percentage of the health assessment.

	Subjective Health Assessment		Total
	Good	Very Good	
underweight	3.00%	3.80%	3.40%
normal body weight	47.00%	55.20%	50.70%
overweight without obesity	33.10%	30.20%	31.80%
I degree of obesity	12.60%	8.60%	10.80%
II degree of obesity	3.30%	1.80%	2.60%
III degree of obesity	1.00%	0.40%	0.70%
Total	100.00%	100.00%	100.00%

**Table 7 jcm-11-03720-t007:** Relationship between BMI and the incidence of selected diseases; data presented as percentage of the BMI category ^1^.

	Underweight	Normal Body Weight	Overweight without Obesity	I Degree of Obesity	II Degree of Obesity	III Degree of Obesity	Total
Hypertension	29.6% _a_	38.4% _b_	45.3% _c_	50.5% _d_	52.3% _e_	55.4% _f_	44.9%
Type 2 diabetes	8.6% _a_	5.8% _b_	6.9% _c_	10.6% _d_	15.6% _e_	18.3% _f_	8.1%
Lipid disorders	58.9% _a_	52.3% _b_	43.9% _c_	35.1% _d_	28.9% _e_	23.8% _f_	43.4%
Coronary disease	2.9% _a,b,c,d_	3.4% _d_	4.0% _c_	3.8% _c_	3.2% _b,d_	2.5% _a_	3.7%
Total	100.0%	100.0%	100.0%	100.0%	100.0%	100.0%	100.0%

^1^ Each letter in subscript represents a subset of the year category whose column proportions do not differ significantly at the level of 5%.

**Table 8 jcm-11-03720-t008:** Relationship between BMI and comorbidities; data presented as percentage of BMI ^1^.

	Underweight	Normal Body Weight	Overweight without Obesity	I Degree of Obesity	II Degree of Obesity	III Degree of Obesity	Total
Hypertension	26.1% _a_	30.5% _b_	33.7% _c_	38.8% _d_	41.0% _e_	45.3% _f_	34.1%
Type 2 diabetes	7.8% _a_	4.1% _b_	2.8% _c_	3.2% _d_	4.6% _b_	5.8% _e_	3.5%
Lipid disorders	57.7% _a_	47.3% _b_	33.0% _c_	19.7% _d_	11.6% _e_	7.5% _f_	33.8%
Coronary disease	2.0% _a_	1.2% _b_	1.0% _c_	0.6% _d_	0.4% _d,e_	0.2% _e_	0.9%
Hypertension + Type 2 diabetes	0.6% _a_	0.9% _a_	2.0% _b_	4.3% _c_	7.7% _d_	10.7% _e_	2.5%
Hypertension + Lipid disorders	4.2% _a_	11.6% _b_	19.5% _c_	21.9% _d_	20.1% _c_	17.0% _e_	17.3%
Hypertension + Coronary disease	0.3% _a_	0.6% _a_	0.9% _b_	1.0% _b_	0.9% _b_	1.1% _b_	0.8%
Type 2 diabetes + Lipid disorders	0.4% _a_	0.7% _a_	1.1% _b_	1.5% _c_	1.8% _d_	1.4% _b,c,d_	1.1%
Type 2 diabetes + Coronary disease	0.1% _a,b,c_	0.0% _c_	0.1% _b_	0.1% _a_	0.2% _a_	0.1% _a,b,c_	0.1%
Lipid disorders + Coronary disease	0.4% _a,b,c_	0.7% _c_	0.7% _c_	0.4% _b_	0.3% _a,b_	0.1% _a_	0.6%
Hypertension + Type 2 diabetes + Lipid disorders	0.2% _a_	0.9% _b_	2.7% _c_	5.0% _d_	8.1% _e_	8.5% _e_	2.9%
Hypertension + Type 2 diabetes + Coronary disease	0.1% _a,b,c_	0.1% _c_	0.2% _b_	0.4% _a_	0.4% _a_	0.4% _a_	0.2%
Hypertension + Lipid disorders + Coronary disease	0.3% _a_	1.2% _b,c_	2.0% _d_	2.0% _d_	1.5% _c_	0.8% _a,b_	1.7%
Type 2 diabetes + Lipid disorders + Coronary disease	0.1% _a,b,c,d_	0.0% _c,d_	0.0% _b,d_	0.1% _a_	0.1% _a,b,c,d_	0.1% _a,b,c,d_	0.1%
All		0.2% _a_	0.6% _b_	1.0% _c_	1.3% _d_	1.1% _c,d_	0.6%
Total	100.0%	100.0%	100.0%	100.0%	100.0%	100.0%	100.0%

^1^ Each letter in subscript represents a subset of the year category whose column proportions do not differ significantly at the level of 5%.

**Table 9 jcm-11-03720-t009:** Relationship between age and comorbidities; data presented as percentage of age group ^1^.

	Age					Total
	<18	18–35	35–54	55–69	>69	
Hypertension	66.7% _a,b,c_	37.7% _c_	34.3% _b_	31.6% _a_	30.2% _a_	34.3%
Type 2 diabetes	33.3% _a_	6.7% _a_	2.8% _b_	2.6% _c_	2.5% _b, c_	3.5%
Lipid disorders		44.3% _d_	35.8% _c_	18.6% _b_	7.1% _a_	33.2%
Coronary disease		0.5% _c_	0.8% _b_	1.8% _a_	2.0% _a_	1.0%
Hypertension + Type 2 diabetes		0.9% _d_	2.1% _c_	4.8% _b_	6.6% _a_	2.6%
Hypertension + Lipid disorders		8.7% _d_	18.3% _c_	22.1% _b_	19.0% _a,c_	17.3%
Hypertension + Coronary disease		0.1% _d_	0.5% _c_	2.1% _b_	4.5% _a_	0.8%
Type 2 diabetes + Lipid disorders		0.6% _c_	1.1% _b_	1.5% _a_	1.4% _a,b_	1.1%
Type 2 diabetes + Coronary disease		0.0% _d_	0.0% _c_	0.2% _b_	0.6% _a_	0.1%
Lipid disorders + Coronary disease		0.1% _c_	0.5% _b_	1.3% _a_	1.9% _a_	0.6%
Hypertension + Type 2 diabetes + Lipid disorders		0.4% _c_	2.2% _b_	6.5% _a_	8.1% _a_	3.0%
Hypertension + Type 2 diabetes + Coronary disease		0.0% _d_	0.1% _c_	0.6% _b_	1.9% _a_	0.2%
Hypertension + Lipid disorders + Coronary disease		0.1% _d_	1.1% _c_	4.3% _b_	9.0% _a_	1.7%
Type 2 diabetes + Lipid disorders + Coronary disease		0.0% _b_	0.0% _b_	0.2% _a_	0.4% _a_	0.1%
All		0.0% _d_	0.2% _c_	1.8% _b_	4.7% _a_	0.6%
Total	100.0%	100.0%	100.0%	100.0%	100.0%	100.0%

^1^ Each letter in subscript represents a subset of the age category whose column proportions do not differ significantly at the level of 5%.

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
