# Peer review of "Prevalence of Obesity and Severe Obesity among Professionally Active Adult Population in Poland and Its Strong Relationship with Cardiovascular Co-Morbidities-POL-O-CARIA 2016–2020 Study"

_jcm, 2022, doi:10.3390/jcm11133720_

Round 1

Reviewer 1 Report

Prevalence of Obesity and Severe Obesity Among Professionally Active Adult Population in Poland and its Strong Relationship with Cardiovascular Co-Morbidities- POL-O-CARIA 2021 Study

In the present study, Rulkiewicz et al. have wonderfully analyzed the large group of data from the POLOCARIA 2016-2020 study (men/women), and they have successfully demonstrated that the intensity of obesity is increased over the time and is associated with many diseases.   

Despite the topic being relevant for obesity and its associated commodities but there are several pitfalls needing further insights

1.     Since the authors analyzed the POLOCARIA 2016-2020 study in the manuscript, they should mention POLACARIA 2016 -2020 in the title as well.

2.  Authors should work on the abstract and be specific as the manuscript is focused on cardiovascular co-morbidities. For example, see page 1, lines 23 and 25. Also Significant level chi2 (70) = 12228,11; p < 0.001 is not clear. Is it in relation to cardiovascular diseases or any other diseases?

3.     In Table 1, the authors have described the relation between BMI and year of measurement beautifully, but it will be better if they also add the SD or SEM and significant level between groups.   

4.     Description of table 2 is not clear. Authors should add more detail in the text about the analysis and significance level between the groups.

5.     In Table 2, the authors described the relationship between BMI, time, and patient sex very clearly, but the statistical analysis is not clear. Authors can either change the format in all tables (add a separate column for statistical analysis) or describe it in the text.

6.  In Figure 1, authors should assign age on Y-axis, not M+ SD because in the text authors have correlated age with BMI. Also, they should add the significant level (P-value) between groups in both text and graph. All the graphs in the manuscript should be clearly visible. It will be good if authors use the GraphPad Prism for graph preparation. Authors should also change the Y-axis (add months) in Figure 2.  

7.     Authors have described the relationship between BMI and smoking in Figure 3 but the significance level and SD between all groups are missing.

8.     Authors have shown the same correlation between BMI and incidents of diseases like hypertension, type 2 diabetes, lipid disorder, and coronary disease in both tables 6 and 7. Then why is there a difference in the incidence of disease percentage among the tables? Also, the author should use the same format in all the tables in the manuscript either a comma (,) or point (.) for the decimals.

9. Authors have analyzed data for cardiovascular comorbidities according to mean BMI from 14-90 years old patients (see table 7). However, it will be good if the authors also add the tables to correlate age distribution with comorbidities because aging is also associated with obesity and its complications.

Author Response

Response to Reviewer 1 Comments

Point 1:

In the present study, Rulkiewicz et al. have wonderfully analyzed the large group of data from the POLOCARIA 2016-2020 study (men/women), and they have successfully demonstrated that the intensity of obesity is increased over the time and is associated with many diseases.   

Despite the topic being relevant for obesity and its associated commodities but there are several pitfalls needing further insights

  1. Since the authors analyzed the POLOCARIA 2016-2020 study in the manuscript, they should mention POLACARIA 2016 -2020 in the title as well.

Response 1: First of all, we thank you very much for your comments that have helped us improve our publication.Thank you for your comment 1. In line with your recommendation and in line with Reviewer 2 recommendation , we have corrected the term and substitute it (both in the title and later ) with POL-O-CARIA 2016-2020 Study:

Point 2:

  1. Authors should work on the abstract and be specific as the manuscript is focused on cardiovascular co-morbidities. For example, see page 1, lines 23 and 25. Also Significant level chi2 (70) = 12228,11; p < 0.001 is not clear. Is it in relation to cardiovascular diseases or any other diseases?

Response 2

Thank you for your comment 2. In line with your recommendation, we corrected our abstract, and trying to be more specific. Details in this respect are provided in Table 7 and it is difficult to summarize them all in abstract. However, we totally agree that these details are highly interesting. 

Abstract: For several decades, a steady increase in the percentage of overweight and obese people has been observed all over the world. There are many studies available in the literature emphasizing the relation of overweight and obesity with the occurrence of other diseases. The aim of the study is to characterize the prevalence of obesity and severe obesity as well as their changes over time among professionally active adults who underwent occupational medicine examinations in Poland in 2016-2020, POL-O-CARIA 2016-2020 Study.. The article analyzes the results of the POLOCARIA 2020 study, concerning adults who are professionally active and visited in the years 2016-2020 as part of occupational medicine. In total, the results of 1,450,455 initial, control and periodic visits as part of the occupational medicine certificate were analyzed. Statistical calculations were performed with the use of IBM SPSS Statistics 25. In both groups (men/women) a significant decrease was observed every year for people who had normal body weight. In addition, the tendency to increase in people with I and III degree obesity was more strongly observed in the group of men. A significant relationship was also observed between BMI categories and the occurrence of all analaysed comorbidities: hypertension, type 2 diabetes, lipid disorders, coronary artery disaease (chi2 (70) = 12228,11; p < 0.001). Detailed results showed that in the group of patients diagnosed with hypertension or lipid disorders, significant differences were observed between all groups - it turned out that as the BMI level increased (I, I, III), the percentage of occurrence of hypertension( 38,1%; 41%; 45,3 % respectively) and type 2 diabetes (3,2%; 4,6%, 5,8% respectively) increased ( p< 0.001) A significant relationship was also observed between BMI categories and the occurrence of comorbidities (chi2 (70) = 12228,11; p < 0.001). Our analysis indicates that the prevalence of adult obesity and severe obesity will continue to increase nationwide, with accompanying large increase in comorbidities.

Point 3:

In Table 1, the authors have described the relation between BMI and year of measurement beautifully, but it will be better if they also add the SD or SEM and significant level between groups.   

Response 3

Thank you for your comment 3. The data in Table 1 present percentages and not the average. For this reason, SD were not added; significance of differences was added (column comparisons with Bonferroni correction)

 Point 4:

Description of table 2 is not clear. Authors should add more detail in the text about the analysis and significance level between the groups.

Response 4: Thank you for your comment 4. In line with your recommendations, we developed/change  description of this table in the test of the article

New description we presented below:

In the case of BMI, it turned out that the longest medical certificates for professionally active Poles were received by people with normal body weight and it equals about 34 months. In the group of people with overweight and obesity, it was observed that, along with the degree of obesity, the average number of months in which the decision was issued decreased significantly (see Table 2.). According to doctor decision, patients with overweight received medical certificate for work for average about 30 months. The ability to work was significantly worse in patients with obesity of the 1st degree ( about 28 months), obesity of the 2nd degree ( about 27 months)  and obesity of the 3rd degree (almost 26 months) – we presented details in table 2.

 Point 5:

In Table 2, the authors described the relationship between BMI, time, and patient sex very clearly, but the statistical analysis is not clear. Authors can either change the format in all tables (add a separate column for statistical analysis) or describe it in the text.

Response 5: Dear Reviewer, We are very sorry, but we think there was a mistake - in the table 2, we do not include data on the sex of patients

Point 6:

In Figure 1, authors should assign age on Y-axis, not M+ SD because in the text authors have correlated age with BMI. Also, they should add the significant level (P-value) between groups in both text and graph. All the graphs in the manuscript should be clearly visible. It will be good if authors use the GraphPad Prism for graph preparation. Authors should also change the Y-axis (add months) in Figure 2.  

Response 6: Thank you for your comment 6. In line with your recommendations, we corrected figure 1 and figure 2 according to your suggestion.

Point 7:

Authors have described the relationship between BMI and smoking in Figure 3 but the significance level and SD between all groups are missing.

Response 7. Thank you for your comment 7. Changed; Figure 3 present percentages and not the average. For this reason, SD were not added;

Point 8:

Authors have shown the same correlation between BMI and incidents of diseases like hypertension, type 2 diabetes, lipid disorder, and coronary disease in both tables 6 and 7. Then why is there a difference in the incidence of disease percentage among the tables? Also, the author should use the same format in all the tables in the manuscript either a comma (,) or point (.) for the decimals.

The data in the two tables differ because Table 7 shows comorbidities. If the data in both tables were percentages to rows (for disorders) then the percentage results would be the same. The larger number of diseases shown in Table 7 explains the difference in percentage results between Tables 6 and 7

Point 9: Authors have analyzed data for cardiovascular comorbidities according to mean BMI from 14-90 years old patients (see table 7). However, it will be good if the authors also add the tables to correlate age distribution with comorbidities because aging is also associated with obesity and its complications.

Response 9. Thank you for your comment 9. In line with your recommendations, we added a new table 8.

Reviewer 2 Report

In the present paper, Rulkiewicz et al. conducted an exploratory analysis on the prevalence of obesity among professionally active adult population in Poland. Authors reported that the percentage of body mass index-based overweight and obese adults depicts an increasing trend over time and relates to the presence of comobordities. The manuscript is timely and highly interesting; however, the following concerns should be addressed:

-          The main hypothesis/research question is missing in the abstract.

-          Is the paper analyzing the results of POL-O-CARIA 2020 or 2021 Study?

-          Have potential confounders been taken into account?

-          The paper lacks methodological details such as how sampling bias was avoided or whether other socio-demographic and lifestyle factors were obtained.

-          It would be highly interesting to correlate being overweight and obesity with socio-economic status.

-          Authors should argue why the number of visits analyzed in 2020 was very low.

-          Since obesity coexists with comorbidities like hypertension or lipid disorders, the relationship with sedentary lifestyle (e.g., by hours spent on screens per day) or physicial activity should be addressed.

-          Due to the retrospective nature of this paper only associations can be inferred.

Minor comments

-          ‘fruits’ instead of ‘friuts’ in page number 3 (line 93).

-          Sentence in page number 3 (lines 117-118) is repeated too often in the manuscript.

-          It should be better use twice reference number 13 rather than reference number 14 being “as above”.

Round 2

Reviewer 1 Report

Prevalence of Obesity and Severe Obesity Among Professionally Active Adult Population in Poland and its Strong Relation- 3 ship with Cardiovascular Co-Morbidities- POL-O-CARIA 2016-2020 Study

In the present study, Rulkiewicz et al. have improved the revised manuscript wonderfully, but still, it has some minor corrections. 

  1. The authors have mentioned the 2016-2-2020 study in the title. They should use only a hyphen between the years, not a number like the 2016-2020 study.  
  2. In figure 3, the format of statistical significance is not looking representative. Authors should move the significance level in the tabular form.
  3. Authors should carefully check all figures, tables, text format, and alignment.  

Author Response

Dear Reviewer,
Please find below all responses. 

Response 1
Thank you for all your comments. They helped id to improve our manuscript. Of course, we corrected our mistake in the title.

Response 2 
Thank you, we corrected figure 3

Response 3
Thank you, we checked our manuscript once again. We corrected some mistakes.